**Observation-derived ice growth curves show patterns and trends in maximum**
**ice thickness and safe travel duration of Alaskan lakes and rivers**
Christopher D. Arp[1], Jessica E. Cherry[1,2], Dana R.N. Brown[3], Allen C. Bondurant[1], and Karen L.
Endres[4]
[1]Water and Environmental Research Center, Unversity of Alaska Fairbanks, Fairbanks, AK
99775, USA
[2]Alaska-Pacific River Forecast Center, National Weather Service, Anchorage, AK 99502, USA
[3]Institute of Arctic Biology, Unversity of Alaska Fairbanks, Fairbanks, AK 99775, USA
[4]Alaska-Pacific River Forecast Center, National Weather Service, Fairbanks, AK 99775, USA
*Correspondence to*: Chris Arp (cdarp@alaska.edu)
**Abstract.** The formation, growth, and decay of freshwater ice on lakes and rivers are
fundamental processes of northern regions with wide ranging implications for socio-ecological
systems. Ice thickness at the end of winter is perhaps the best integration of cold-season weather
and climate, while the duration of thick and growing ice cover is a useful indicator for the winter
travel and recreation season. Both maximum ice thickness (MIT) and ice travel duration (ITD)
can be estimated from temperature-driven ice growth curves fit to ice thickness observations. We
simulated and analyzed ice growth curves based on ice thickness data collected from a range of
observation programs throughout Alaska spanning the past 20 – 60 years to understand patterns
and trends in lake and river ice. Results suggest reductions in MIT (thinning) in several northern,
interior, and coastal regions of Alaska and overall greater interannual variability in rivers
compared to lakes. Interior regions generally showed less variability in MIT and even slightly
increasing trends in at least one river site. Average ITD ranged from 214 days in the northern-
most lakes to 114 days across southern-most lakes with significant decreases in duration for half
of sites. River ITD showed low regional variability, but high interannual variability,

underscoring the challenges with predicting seasonally-consistent river travel. Standardization and analysis of these ice observation data provide a comprehensive summary for understanding changes in winter climate and its impact on freshwater ice services.

## 1 Introduction

Arctic amplification is an enhanced warming response in high latitudes relative to increasing global temperature (Serreze and Barry 2011). Though not yet completely understood, sea ice decline and associated climate feedbacks are considered to be major drivers of this process (Serreze and Francis 2006). A salient feature of arctic amplification is greater warming during winter, which has been strikingly apparent in Alaska during recent years (Wendler et al. 2014, Walsh and Brettschneider 2019). Terrestrial landscape responses to winter climate change are perhaps most quantifiable in ice formation and growth on lakes and rivers, and readily described by ice thickening through the winter (Allen 1977, Engram et al. 2018). Freshwater ice thickness and duration  may function as robust integrators of winter climate, as they respond both to changes in air temperature and snow accumulation. Additionally, ice thickness and its duration have important implications for winter travel, subsistence, and recreation in Alaska and across the Arctic (Brown and Duguay 2010, Schneider et al. 2013, Cold et al. 2020).

Some of the longest ice thickness records come from the Barrow Peninsula in northern-most Alaska where lake ice historically grew greater than 2-m thick in some years by winter's end as recorded in the 1970s (Weeks et al. 1978) and 1990s (Zhang and Jeffries 2000). In recent years however, MIT did not exceed 1.2-m in snowy winters, when ice was well insulated and air temperatures were unusually warm (Arp et al. 2018). The impacts of arctic amplification on

freshwater ice should be most evident in this northern coastal region where Alexeev et al. (2016)

demonstrated a linkage between sea ice extent and lake ice growth. This long, yet intermittent,

record of lake ice thickness in northern Alaska comes from a variety of observation efforts

including community-based monitoring facilitated by government agencies (Bilello 1980) and

school science programs (Morris and Jeffries 2010). These same community-based monitoring

programs also contributed to shorter, but still highly valuable, ice thickness records for other

lakes and rivers in Alaska, which have been maintained and extended by the National Weather

Service's Alaska Pacific River Forecast Center (APRFC). APRFC's interest in ice thickness has

primarily been to facilitate river breakup and ice jam forecasting, but is also of value for

informing safe ice travel, as fall through accidents have increased in recent years (Fleischer et al.

2014).

In contrast to ice thickness observations, records and analysis of ice phenology (timing of

freeze-up and break-up) are often long and abundant for many northern regions, likely owing to

the ease of observing water-to-ice transition timing from shorelines, aircraft, or satellites (Brown

and Duguay 2010). Rigorous satellite-based observations show distinct trends towards earlier

break-up on both rivers (Cooley and Palvesky 2016) and lakes (Smejkalova et al. 2017) in

Alaska and longer-term observer records from other northern regions show similar patterns (e.g.,

Magnuson et al. 2000, Weyhenmeyer et al. 2011). Detection of trends in freeze-up timing are

often less certain (Brown and Duguay 2010), though recent analyses suggest late freeze-up

contributions to reduced ice cover duration on lakes (Sharma et al. 2019) and rivers (Yang et al.

2019). Brown et al. (2018) tracked both freeze-up and break-up progression in Alaskan rivers,

highlighting the varying stages of ice formation and decay processes relative to access for over

ice travel, and suggests the need to move beyond ice phenology as an exact *to-the-day* event.

Tracking changes in ice thickness through the winter cold season provides the simplest means of
quantifying this continuum, though these data are distinctly more field intensive.

3         Reported analysis of ice thickness datasets often lack detection of thinning trends despite

progressive winter warming, which may be due to high interannual variability in snowfall and
the dominant role of snow insulation on ice growth (Brown and Duguay 2010). Yet the majority
of studies analyzing ice thickness trends that we are aware of came before unprecedented warm
winters during the last decade in Alaska (Wendler et al. 2014, Walsh and Brettschneider 2019).
We also suspect that the inherent nature of ice thickness data collection, in which the timing of
late winter measurement may vary from year to year relative to slight shifts in the ice growth
season, adds additional noise in detecting trends. Other factors related to ice observations include
that ice thickness can vary significantly within a small area depending on snowcover,
measurement protocols have often differed among programs, and rural community observers are
often volunteers with high turnover and minimal oversight.  Large data gaps in records also make
it difficult to ascertain trends at some long-term sites (Cherry 2019). For these reasons, we are
motivated to standardize ice thickness according to ice growth curves informed by field
observations and calculate relevant metrics, Maximum Ice Thickness (MIT) and Ice Travel
Duration (ITD), as well as to merge analyses of both river and lake ice in Alaska.

18        In this study, we organized winter lake and river ice thickness observations from a variety

of sources to provide an updated analysis of patterns and trends of ice growth in Alaska from
1962 to present. Fitting these data to air temperature-driven ice growth curves simulated with the
Stefan equation (Stefan 1891, Jumikis 1977) provides a robust seasonal estimation of changing
ice thickness for multiple sites with proximate climate data. The ice growth curves were used to
estimate MIT and ITD for four lakes and four rivers distributed across Alaska, with records
spanning 20 years or greater, to provide a summary of recent changes in ice of climatic and
societal relevance.  Several shorter records are also presented for spatial comparison.
**2 Background and Methods**
**2.1 Study region and waterbodies**
The diverse geographic and northern climatic setting of the State of Alaska presents a fascinating
study region to observe freshwater ice (Arp and Jones 2009, Arp et al. 2013). Even though the
state of Alaska is a geopolitical unit, its vast size (1.5 million $km^2$), expansive latitudinal extent
(18°), wide longitudinal extent (58°), lengthy coastline (54,500 km), and complex tectonic
setting create a largely contiguous landscape with several large mountain ranges and expansive
river valleys (Figure 1). These geographical attributes of Alaska interact with climate, glacial
history, and soil conditions (particularly permafrost) to create many lakes (> 400,000) and
extensive and varied river networks (>150,000 km) (Arp and Jones 2009).  In contrast to
waterbody extents, the Alaska road network is relatively short (<25,000 km) and not connected
to the majority of towns and villages, which limit opportunities to maintain long-term
observations of most waterbodies. Thus the majority of sites with long-term ice observation data
are associated with large towns along roadways or villages adjacent to rivers or lakes (Table 1).
Another restriction for this study was proximity to reliable long-term air temperature data from
weather stations, which are typically associated with larger airports.

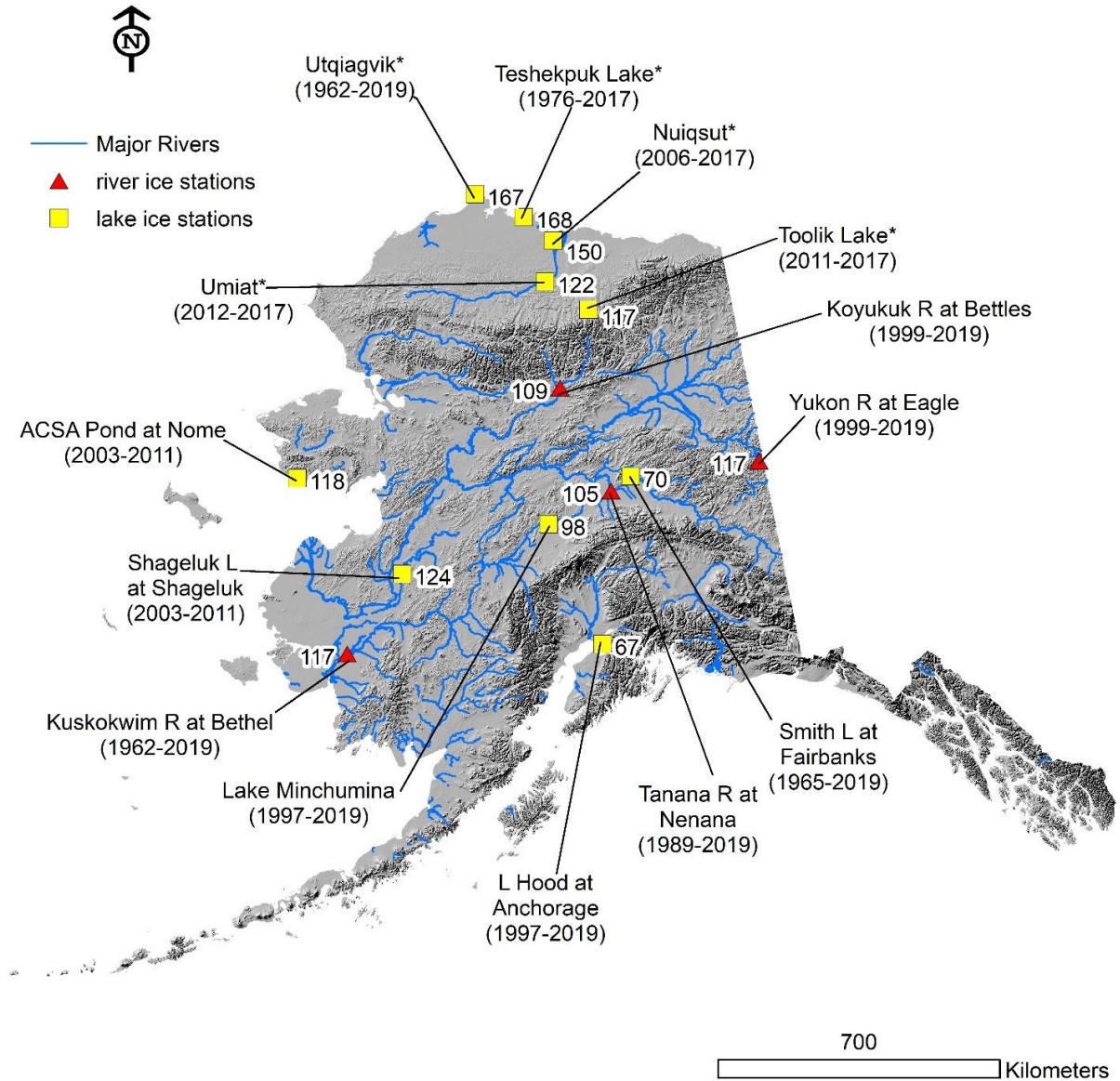

**Figure 1**. Map of the State of Alaska with all observation locations by waterbody and nearest
community indicated and average maximum ice thickness (cm) for each period of record (in
parenthesis). Several shorter term records (<20 years) are shown here for additional context
which are not presented in the main analysis (* indicates that observations are based on multiple
lakes or rivers within a region; 300 m DEM hillshade is USGS data in the U.S. public domain).

**Table 1.** Summary of major ice observation stations and corresponding ice thickness records. Ice growth curves simulated using the Stefan equation and the average α coefficient (cm $^0$ C$^{-1/2}$ d$^{-1/2}$) and accumulated freezing degree days (AFDD) at MIT ($^0$ C d) are reported for each station. Air temperature data from National Weather Service stations are indicated by station codes.

| Water Body | Community | Region | Period | Years Observed | α (ave) | AFDD (ave) | Weather Station |
|---|---|---|---|---|---|---|---|
| multiple lakes | Utiqagvik | North Slope | 1962-2019 | 26 | 2.6 | -4166 | Barrow-PABR |
| Koyukuk River | Bettles | Interior | 1968-2019 | 48 | 1.9 | -3265 | Bettles-PABT |
| Yukon River | Eagle | Interior | 1999-2019 | 21 | 2.4 | -2594 | Eagle-PAEG |
| Smith Lake | Fairbanks | Interior | 1965-2019 | 53 | 1.3 | -2895 | Fairbanks-PAFA |
| Tanana River | Nenana | Interior | 1989-2019 | 31 | 2.2 | -2516 | Nenana-PANN* |
| Lake Minchumina | Minchumina | Interior | 1997-2019 | 18 | 2.0 | -2542 | Nenana-PANN, McGrath-PAMC |
| Lake Hood | Anchorage | South-Central | 1997-2019 | 22 | 2.6 | -921 | Anchorage-PANC |
| Kuskokwim River | Bethel | Western | 1962-2019 | 48 | 3.3 | -1826 | Bethel-PABE |

*Missing data derived from relationship to Fairbanks-PAFA.

## 2.2 Alaska ice observation programs

Scientific records of ice thickness measurements in Alaska date back to the 1[st] International Polar Year in 1882-83 for lakes located near Barrow (Ray 1885). Starting in the early 1960's the U.S. Army Corps of Engineer's Cold Regions Research and Engineering Laboratory (CRREL) established ice observing stations in coordination with Canadian and U.S. government agencies (Bilello 1980). Stations included 26 lakes and rivers in Alaska where ice thickness data were collected at weekly intervals until at least 1974. Observations made by local residents (Alaska Natives, homesteaders, lodgekeepers, teachers, and clergy) for up to 15 years, in some cases, provided valuable data for developing ice growth and decay models as reported in Bilello (1980) and other CRREL reports. Perhaps as importantly, these data provide a comprehensive summary

of ice thickness and its variability over climatically diverse region of the Arctic and sub-Arctic

before notable climate warming. The majority of Alaska ice thickness data, as well as snow

depth data, from this program are now archived with the Arctic Data Center (ADC) (Bilello

2019). Observation protocols are not always well described in the reports for this program, but

likely relied on narrow gauge hand-augers and tapes similar to current procedures, and the ice

thickness observations per date may have come from single point measurements.

A more recent winter observation program, Alaska Snow and Ice Observation Network

(ALISON), pioneered the integration of science education with snow and ice physics in Alaska

schools from 1999 to 2010 (Morris and Jeffries 2010). While learning about snow and ice

physics, students and teachers collected valuable datasets across a wide range geographic and

climatic settings, including at least 17 lakes ranging from the Barrow Peninsula in the north to

the Kenai Peninsula in the south. Several of these sites overlap with CRREL stations, thus

providing an opportunity for extension of records and temporal comparisons. ALISON's focus

on snow depth, density, and heat flux provided additional data of value for understanding

variability and modeling ice growth (Gould et al. 2005, Jeffries et al. 2005). ALISON datasets

are also archived at ADC (Morris and Jeffries 2019). Often up to 20 snow measurements were

recorded per sampling interval for this program, while ice thickness was typically recorded from

a single point using a thermal resistance (heated) wire known as a TWIT (Thermal Wire Ice-

thickness Thingy) (M. Jeffries personal communications). The TWIT was designed to minimize

local snow disturbance that would affect subsequent measurements, but also represents a single

point measurement of ice thickness.

An important Alaska ice record focused on breakup timing of a single river reach is the

Nenana Ice Classic (NIC), where a tripod is set on the Tanana River by the community of

Nenana each year and cabled to a clock to record the exact data of river breakup (Sagarin and

Micheli 2001). This community-based monitoring program dates back to 1917 and each year

thousands of people submit breakup guesses into a pool with the closest participants taking home

>$300,000 US dollars in recent years. Regular ice thickness observations, dating back to at least

1989, have been made by community members who run the NIC, and are published to provide

contestants with additional information to aid in guesses.

A fourth and also contemporary Alaska-wide lake and river data source is provided by

the NWS Alaska-Pacific River Forecast Center (APRFC). Prior to the establishment of the

APRFC, the National Weather Services weather forecast offices collected or solicited these data

and maintained them at what is now the National Center for Environmental Information. Historic

monthly ice thickness measurements have been compiled from a variety of sources including the

CRREL dataset (Bilello 1980) and contemporary observation come from NWS scientists and

paid and volunteer observers in remote communities throughout the state. Much of this ice

thickness data for both rivers and lakes is used in operational forecasting of river conditions

specific to river break-up and ice jam flooding predictions. APRFC ice thickness data are

available online (https://www.weather.gov/aprfc/IceThickness). Current protocol for APRFC are to

collect single-auger hole and tape observations near the start of the month from November to

March near the same location at each waterbody below undisturbed snow.

Lastly, a regional lake ice observation program focused on Alaska's North Slope began in

2012 called the CircumArctic Lakes Observation Network (CALON) (Hinkel et al. 2012). This

project supported by the National Science Foundation collected consistent late winter ice

thickness data from sets of six lakes in each of ten study areas arrayed from the Brooks Range

foothills to the Beaufort Sea coastal plain until 2017. Several of these study areas were

associated with field camps or other long-term research locations where prior lake ice data
existed or was expected to continue after CALON concluded. Observations were made at the
same location on each lake every year by drilling 3-5 holes at regular spacing and recording
snow data in association with individual ice thickness measurements. ADC has CALON datasets
archived for ice thickness (Arp 2018a) and snow characteristics (Arp 2018b) separately.

6        Addressing the issue of data comparability among programs is relevant to this study. It is

of course expected that higher numbers of samples per site correspond to more accurate ice
thickness measurements such that CALON protocols have higher accuracy than some of the
other ice observation programs described. Analysis conducted in a 36-member sampling protocol
on two lakes and two rivers (one each in the Interior and the North Slope) showed that more
samples reduced error (Figure 2). This analysis suggests that making one observation (n=1)
results in potential error up to 18, 10, 8, and 4 cm from the mean for Interior rivers, North Slope
rivers, North Slope lakes, and Interior lakes, respectively. Three observations (n=3) results in
potential error up to 9, 6, 5, and 3 cm from the mean for Interior rivers, North Slope rivers, North
Slope lakes, and Interior lakes, respectively. While this analysis provides guidance for
comparing the quality of differing ice observation approaches, we also suggest that professional
experience or local knowledge in selecting locations of representative ice thickness may well
overcome low sample size in many cases. Our expectation is that observers reporting to APRFC
have such experience, as do observers from previous programs.

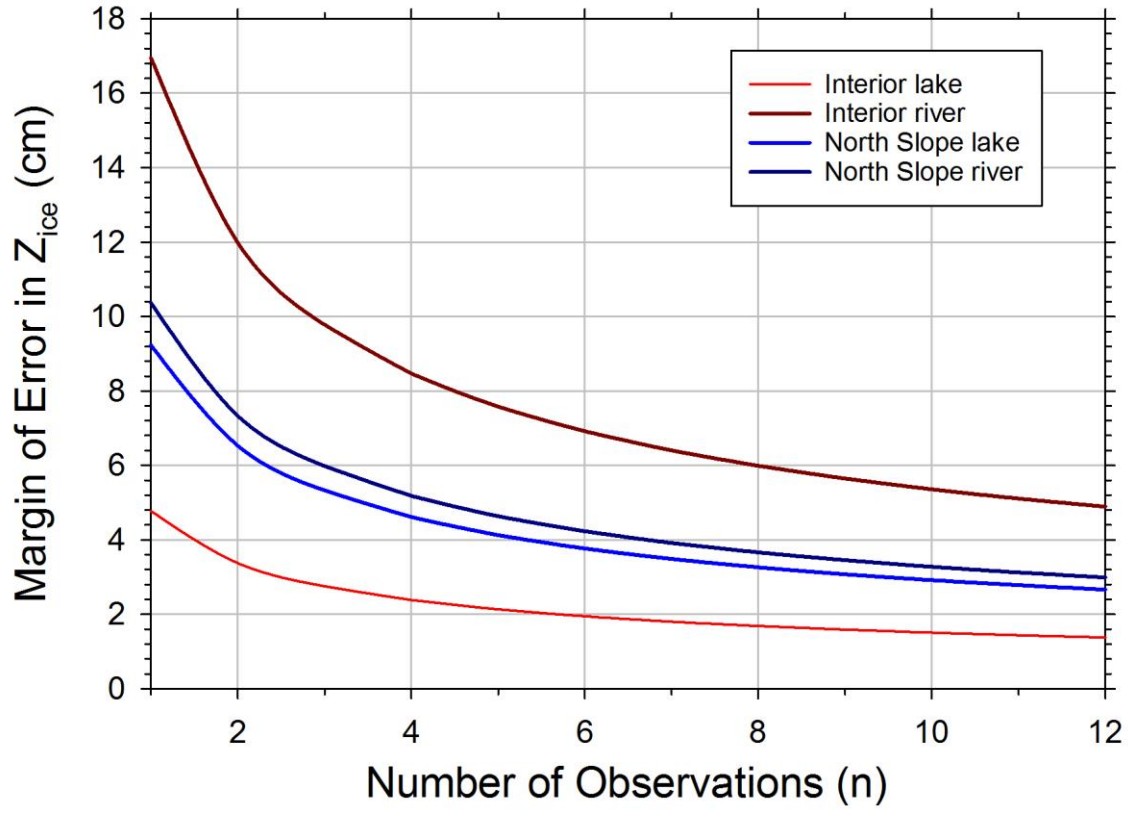

**Figure 2**. Margin of error analysis for Interior and North Slope lakes and rivers from samples of
36 evenly spaced measurements during the winter of 2018-19.

**2.3 Ice growth curve simulation**
For all records with late winter ice observations, we estimated MIT using the Stefan ice growth
model (modified Stefan equation) (1):

$$Z_{ice} = \alpha\sqrt{AFDD} \tag{1}$$

where $\alpha$ is the ice growth (thermal insulation or heat exchange) coefficient in cm $^{0}$ C$^{-1/2}$ d$^{-1/2}$ and
AFDD is accumulated freezing degree days in $^{0}$ C d are used to estimate ice thickness ($Z_{ice}$) in
cm on a daily time-step (Stefan 1891, Jumikis 1977). Mean daily air temperature data used to
force this model was acquired from the nearest National Weather Service (NWS) station, the
majority of which were within 20 km of the water body where ice was monitored (Table 1). In
the cases of Shageluk and Lake Minchumina (Figure 1), no adjacent NWS or other
meteorological stations air temperature records were available and we averaged the records
between two nearest stations set in opposing directions. Summation of AFDD was started at the
date of estimated ice growth initiation each winter. When early ice thickness observations (i.e.
$Z_{ice} < 10$ cm) were available or actual observation of the day of ice initiation, these data guided
selecting this date. When these data were not available, the more common case, we selected the
date according to three consecutive days with mean daily temperature $< 0^0$ C for lakes, which is
based on previous camera and sensor observations (Arp et al. 2013) and six consecutive days
$<0^0$ C for rivers. The later criteria for rivers is based on information from APRFC observers, but
is more limited and thus considered much more uncertain and a potential source of error in ice
growth modeling on rivers. Both air temperature criteria follow guidance in Bilello et al. (1964)
and Ashton (1989) on ice cover initiation and early growth. Complementing the ice growth
model (1), is an ice decay model developed by Bilello (1980) (2):
$$Z_{ice} = \alpha' ATDD \qquad\qquad (2)$$
where α' is the ice decay coefficient in cm $^0$ C$^{-1}$ d$^{-1}$ and ATDD is accumulated thawing degree
days in $^0$ C d with a 0 $^0$ C and also with summation beginning at the day of ice growth initiation.
Equation 2 is calculated concurrently with equation 1 to estimate the ice growth-decay curve
(Figure 3) for the winter season in lakes and rivers. The ice thickness curve was then fit to late
winter $Z_{ice}$ observations for winter season primarily by adjusting α. In some cases when
observation came after the maximum in ATDD, α' was also adjusted to provide the best fit,
otherwise α' was left constant (typically set at 1.0) among simulations with no effect on
estimations of MIT or ITD. MIT was then extracted from each record as shown in the example
ice growth curve in Figure 3. All subsequent data analysis use the MIT as a standardized
estimate of winter ice growth for each winter season and waterbody. Individual year estimates of
MIT and the original ice thickness observations and corresponding model parameters will be
archived and available at the ADC (Arp and Cherry 2020).

6        A new metric was also derived from each ice growth curve, which we term Ice Travel

Duration (ITD) or Safe Travel Duration, and is intended to represent the period of time when
most modes of common travel are safe according to ice thickness and continued thickening.
Quantitatively this is defined as the date when ice thickness surpasses 30 cm to the date of MIT,
the latter of which typically corresponds to the maximum AFDD and the start of ice decay
(Figure 3). Our rationale is that 30 cm exceeds the thickness when most vehicle travel is safe on
freshwater ice. After MIT is reached and ice decay begins, even though its thickness typically
well exceed 30 cm, its structural integrity and strength is changing rapidly such that thickness is
less relevant to its load bearing capacity (Gold 1971, Leppäranta 2015).  In practice it is common
for safe foot and snowmachine travel on thicknesses less than 30 cm and similar travel is
common over thick ice that is starting to degrade. We also note that modeling ice growth during
the initial thickening phase is less predictable by air temperature (Ashton 1989) and thus selected
a level of thickness where we expect this relationship to be more robust. To evaluate how closely
this ITD start date tracked ice conditions identified by local APRFC observers, we compared
"Safe for Vehicle" dates on three rivers and one lake common to our dataset when
"snowmachine" was indicated for "Type of Vehicle" in the APRFC database
(https://www.weather.gov/aprfc/freezeUp) (Figure 4a). This comparison showed a close match
with an average offset ranging from +4 days on the Yukon River at Eagle to -5 days on Lake
Minchumina. Similar comparison was made with the ITD end date tracked observer data
(https://www.weather.gov/aprfc/breakupDB) (Figure 4b). For this comparison in all but two
cases, observer indications of "Safe for Vehicles" was always later than the date of maximum ice
thickness, as we would have anticipated because it is common for travelers to use degrading ice
safely for some period before complete breakup.  Thus, our estimates of ITD should be
considered conservative and for many modes of travel (and corresponding levels of caution) our
estimates of ITD are shorter in duration that what is practiced locally.

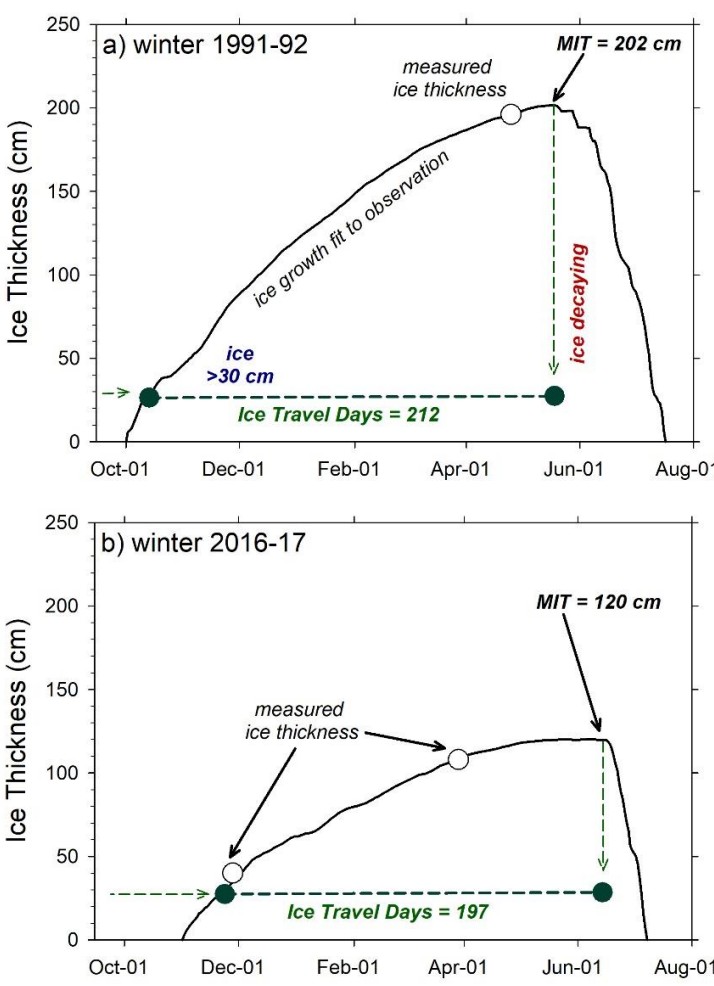

**Figure 3.** Example ice growth and decay curves from Barrow lakes in a thick (a) and thin (b) ice
year showing curve fits to observed data, the time when maximum ice thickness (MIT) is
reached, and the period representing ice travel duration (ITD).

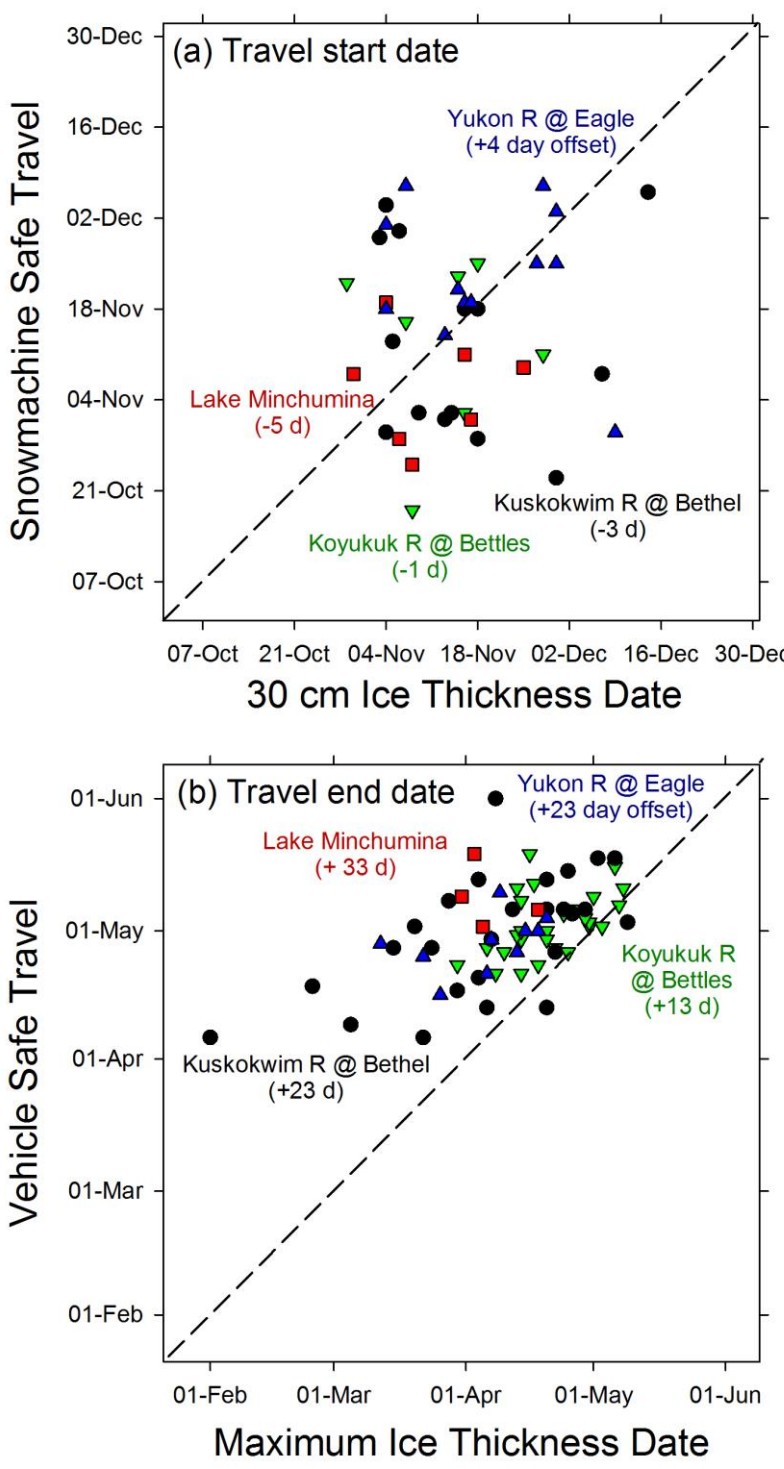

**Figure 4.** Comparison of Ice Travel Duration metric start (a) and end (b) dates for four study
rivers and lakes to ice condition designations reported by Alaska Pacific River Forecast Center
(APRFC) observers.

**2.5 Data analysis**
Patterns and trends in MIT and ITD records for each station were primarily analyzed graphically
in time-series by comparison among lake and river sites. Detection of trends was done using
linear regression for the entire record available for each waterbody. To separate or break MIT
records into distinct periods of potential interest, we used a combination of piece-wise linear
regression (Systat 2013) to identify significant changes in trends and regime shift detection
methods (Rodionov 2004) to identify significant changes in means and variance. Gaps in
observational data also resulted in gaps in MIT estimates, such that many of the breaks or record
separations graphically selected correspond to these missing years of record. We report
significant trends ($p<0.05$) and the means and standard deviations of MIT and ITD for all periods
within the record and this served as our primary basis for comparison and analysis. Multiple
regression analysis was used to evaluate relationships between MIT and ITD to air temperature
and upland snow depth data from proximate weather stations when available to evaluate general
controls on interannual variability.
To understand the relative contributions of thermal forcing (air temperature) and thermal
resistance (primarily snow insulation), we isolated these components from the Stefan equation
(1) over each MIT record per waterbody using power law analysis. In this approach thermal
forcing is represented by AFDD (3),

$$\sqrt{AFDD} = a\, MIT^b \tag{3}$$

and thermal resistance is represented by α (4),

$$\alpha = c\, MIT^d \tag{4}$$

where the coefficients b and d represent the proportions of interannual variation in MIT
explained by the AFDD and α, respectively, and should sum to 1 (b + d = 1). The additional
check that this partitioning of variability follows a power law is that the product of the
coefficients a and c are 1 (a x c = 1). This approach is borrowed from hydraulic geometry
analysis of channels to determine the relative contributions of changing width, depth, and
velocity on discharge. An almost identical result is obtained using multiple regression analysis
on the same variables when comparing the partitioned sum of squares to the total sum of squares
for AFDD and α. We used this power law analysis instead because it seems mathematically and
graphically simpler. For several records, power law coefficients did not balance as expected, and
this was also evident from non-significant model fits for equation 3, 4, or both, and in these cases
were not able to partition the balance of thermal forcing and thermal resistance.
**3 Results**
**3.1 Patterns and trends in lake ice**
Latitudinal patterns in ice thickness exist in Alaska, yet location relative to mountain ranges,
river valleys, and coasts, with varying degrees of sea ice influence, likely had a larger influence.
Spatial patterns of MIT averaged over periods greater than two decades ranged from 67 cm in
Southcentral (Anchorage) and 70 cm in the Interior (Fairbanks) to 167 cm on the Arctic Coastal
Plain of northern Alaska (Utqiagvik) (Table 2). Intermediate average MIT ranged from 98 to 122
cm from shorter term records collected within the last two decades in western Alaska, along the
Alaska Range separating the Southcentral Region from the Interior, and the Brooks Range
foothills of the North Slope (Figure 1).

**Table 2**. Summary of maximum ice thickness (MIT) and ice travel duration (ITD) according to the mean, minimum, and maximum values for each period of record reported here by nearest community to observed lakes (blue) and rivers (green). Years corresponding to minimums and maximums are in parentheses and metrics with significant trends are in **bold**.

| *Metric* | Utiqagvik | Bettles | Eagle | Fair-banks | Nenana | Minchu-mina | Anch-orage | Bethel |
|---|---|---|---|---|---|---|---|---|
| Ave MIT (cm) | **167** | 109 | 117 | 70 | 105 | **98** | 67 | 117 |
| Min MIT | 114 | 69 | 67 | 53 | 69 | 62 | 46 | 69 |
| *(year)* | *(2018)* | *(2009)* | *(2004)* | *(1998)* | *(2018)* | *(2014)* | *(2003)* | *(2019)* |
| Max MIT | 211 | 182 | 157 | 101 | 137 | 123 | 84 | 159 |
| *(year)* | *(1970)* | *(2013)* | *(2015)* | *(1971)* | *(1994)* | *(1997)* | *(2011)* | *(1971)* |
| Ave MIT Date | 29-May | 18-Apr | 5-Apr | 5-Apr | 3-Apr | 6-Apr | 20-Mar | 2-Apr |
| Ave ITD (days) | **215** | 157 | 138 | 135 | 135 | **144** | 110 | 134 |
| Min ITD | 192 | 109 | 101 | 105 | 98 | 108 | 31 | 79 |
| *(year)* | *(2013)* | *(1994)* | *(2019)* | *(1980)* | *(2019)* | *(2019)* | *(2003)* | *(2019)* |
| Max ITD | 239 | 191 | 180 | 165 | 155 | 170 | 158 | 180 |
| *(year)* | *(1971)* | *(2013)* | *(2013)* | *(2013)* | *(2013)* | *(2013)* | *(2002)* | *(2013)* |

Overall the thickest ice and the steepest long-term thinning trend came from lakes on the

Barrow Peninsula (Figure 5a), set between the Chukchi and Beaufort seas (Figure 1). A

significant decrease in MIT from 1962 to 2019 of 0.9 cm/yr is most prominent in the period of

continuous record between 2003 and 2019 with 2.9 cm/yr of thinning ($r^2$=0.35, p=0.01) (Figure

5a). Analyzed separately, the earlier period with more intermittent observations showed no trend

and an average thickness of 177 cm. In comparison, MIT for four of the past six years was less

than 121 cm—a thickness not reported once during the previous 22 years with observations

dating back to 1962. MIT on Barrow Peninsula lakes was typically reached between late May

and early June and average number of safe travel days was 215 with a significant decline of 0.5

d/yr ($r^2$=0.54, p<0.01) (Figure 6). Air temperature during the ice growth season averaged -17.9

$^0$ C with an increasing trend over the period of analysis and with temperature above -15 $^0$ C

during the last four of six years. Upland snow depth averaged 22 cm with no trend. Winter
temperature during the ice growth season explained the majority of variation in MIT for this
record ($r^2$=0.61, p<0.01) and upland snow depth was poorly correlated.

4        The next longest and nearly complete record comes from Smith Lake in Fairbanks dating

back to 1965 (Figure 1). Interestingly, no overall trend was observed over this 54-year period,
though the two thickest MIT years were in 1971 and 1977. A thinning trend of 1.1 cm/yr
($r^2$=0.27, p=0.02) was noted during the first 19 years, while the middle period (1986-2003) had
an average MIT of 72 cm and the last period had average MIT of 66 cm with no trend (Figure
5b). Very thin MIT, less than 60 cm, occurred in all three periods, and MIT exceeding 80 cm
occurred as recently as 2017. Smith Lake is a shallow thermokarst lake with wind-protecting
forest around its entire perimeter, typically with relatively deep uniform snowcover.
Contemporary synoptic comparisons made by APRFC on several larger and less protected
interior lakes within a 100 km radius often show Smith Lake having the thinner ice regionally.
Still, Smith Lake may be widely representative of the many small thermokarst lakes and pond
surrounded by forest in interior Alaska. MIT was typically reached by early April and average
ITD was 135 days with no trend over time (Table 2). Air temperature during the ice growth
season averaged -16.8 $^0$ C with a slight increasing trend and upland snow depth averaged 36 cm
with no trend. Neither upland snow depth nor air temperature during the ice growth season
explained interannual variation in MIT or ITD for this lake.

20       In contrast, the larger and more southerly, but slightly higher elevation Lake Minchumina

(Figure 1) had a MIT of 98 cm averaged over 18 years of mostly continuous observation. MIT of
this lake became notably thinner over this period, decreasing 1.6 cm/yr ($r^2$=0.28, p=0.02) (Figure
5c). Yet, nearly the thickest MIT in this record, 121 cm, was recent, 2013, and then the thinnest
MIT of 62 cm was also recent, 2014 (Table 2). MIT was typically reached by early April and
average ITD was 144 days with a relatively steep declining trend of 1.5 d/yr ($r^2$=0.33, p<0.02)
over this short period (Figure 6). We estimated that the most recent year of record 2019 had the
shortest ITD of 108 days, while another relatively recent year 2013 had the longest ITD of 170
days (Table 2). The long and relatively cold ice growth season of 2013 was also prominent in
other records of ITD in interior sites. Air temperature during the ice-growth season averaged -
14.8 $^0$ C (between Nenana and McGrath stations) with no trend detected.

8        The most southerly site, Lake Hood, located near the Anchorage International Airport

and used as a major floatplane runway, had a similar duration record as Lake Minchumina from
1997 to 2019. Observations on this lake began in 1967, but these appeared anomalously thick,
with many exceeding 140 cm and α coefficients averaging 4.8. We suspect that observers may
have collected measurements along areas where ski-planes or snow grooming compacted the
snow significantly to allow this level of thickening with modest winter temperatures. For the
period when α coefficients are within a more normal range (1.6-3.7) (Table 1), a non-significant
increase of 0.3 cm/yr was detected with an average MIT of 67 cm (Figure 4d). MIT was typically
reached by late March and average ITD was 110 days with lots of variation, but no trend over
this period (Figure 5). We estimated a very short ITD of 31 days over the winter of 2002-03
(Table 2) when ice growth only surpassed 30 cm by 31-Dec and MIT was reached early, 2-Feb.
That winter, as well as 2007-08, had numerous freeze-thaw periods through the normal ice
growth season corresponding to short estimates of ITD (Figure 6). Ice-growth season
temperature averaged -6.4 $^0$ C and upland snow depth averaged 29 cm, with no trends detected.
We suspect snow on this lake would be greatly reduced in some years due to wind-scour along
with varying alteration from human activities.

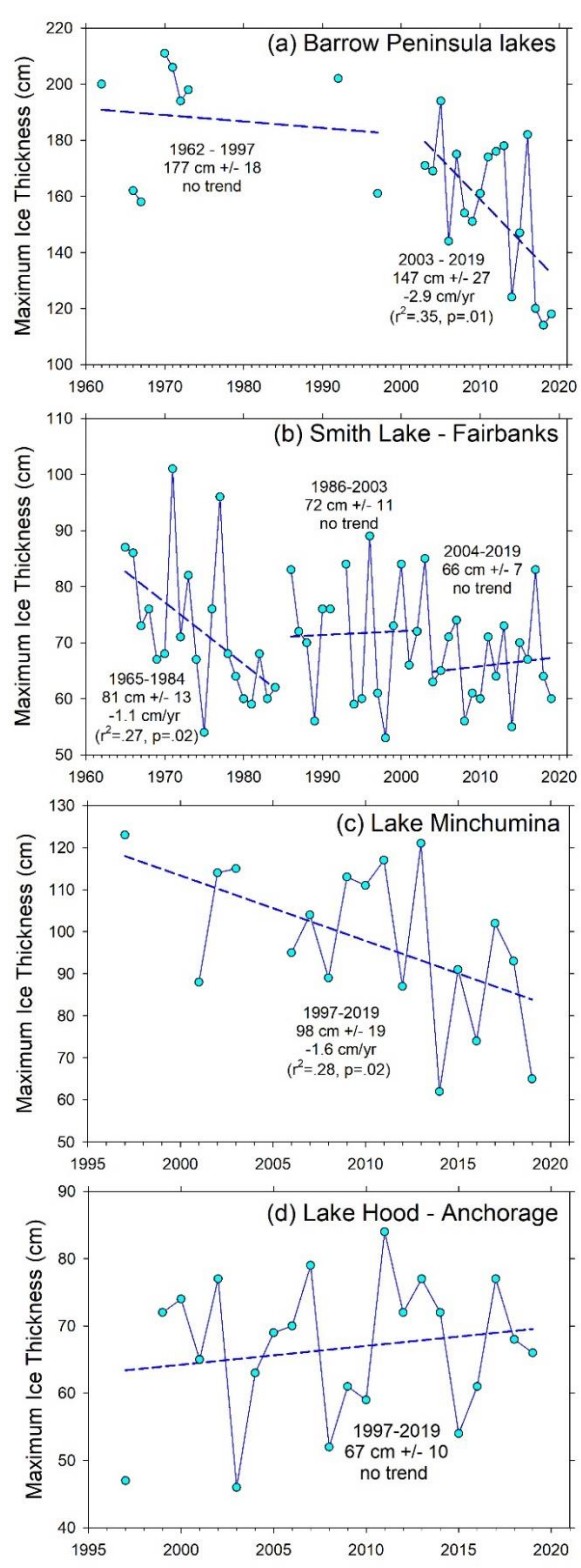

**Figure 5**. Maximum ice thickness patterns and trends for lakes around the Barrow Peninsula (a),
Fairbanks (b), Minchumina (c), and Anchorage (d) for each station's period of record.

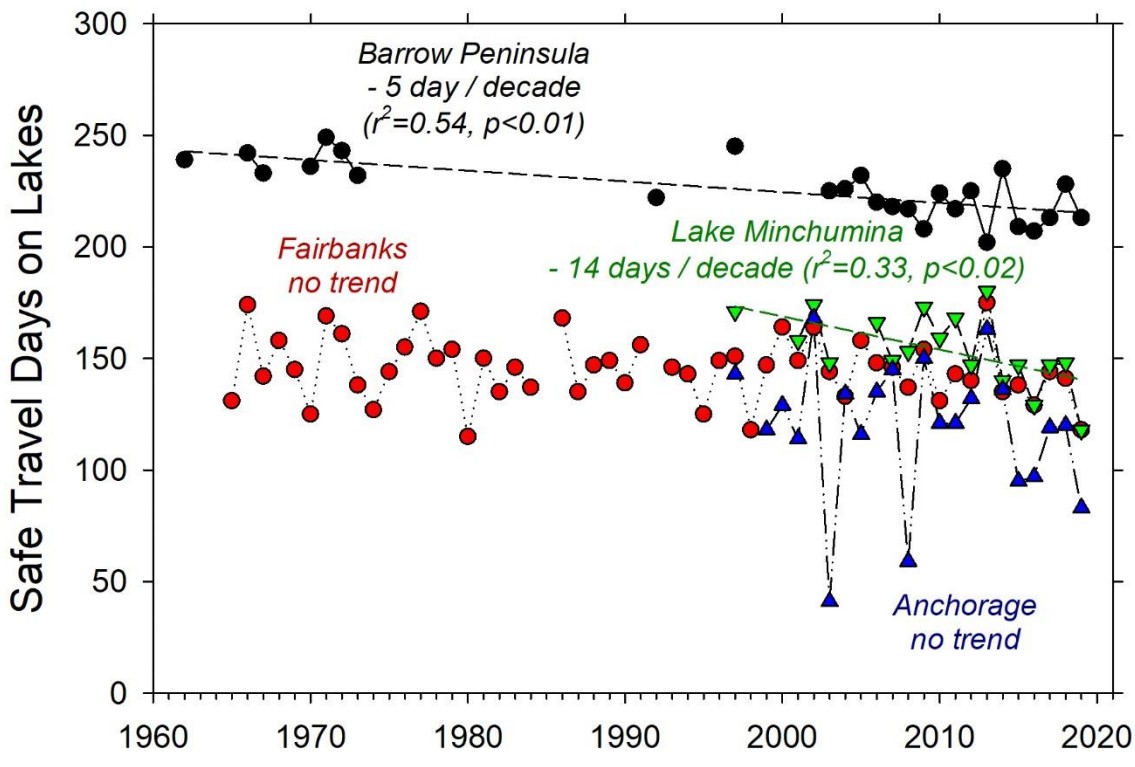

**Figure 6**. Ice travel duration on lakes estimated from ice growth curves for each period of
record.
**3.2 Patterns and trends in river ice**
In contrast to lake ice records, fewer river ice thickness records were analyzed in this study. All
of these observation sites are on larger rivers within the Yukon and Kuskokwim drainages
(Figure 1), associated with river-side communities. A primary use of these ice thickness data is
in forecasting ice jam floods and safe travel conditions. Average MIT over these differing
periods of observation ranged from 98 cm on the Tanana River at Nenana (40 years) in the
Interior to 117 cm at both the Kuskokwim River at Bethel (57 years) near the Bering Sea and
Yukon River at Eagle (20 years) near the Canadian border (Figure 1).
Ice observations on the Kuskokwim River began with the CRREL program in 1962 with
a decade-long gap starting in 1985 and continued again in 1996 with measurements reported to
APRFC. The earlier period (1962-1984) had average MIT of 124 cm with the thickest ice of 159
cm recorded in 1997, no estimates of MIT less than 100 cm, and no trend (Figure 7a). The
second period running to the present also showed no trend, but had somewhat lower average
MIT of 111 cm and higher interannual variation than the first period.  The thinnest MIT, 69 cm,
occurred in 2019 and also corresponded to the earliest river breakup on record, 31-March. Also
in contrast to the first period, MIT was less than 100 cm during eight of 24 winters in the recent
period (Figure 7a). However, thick ice (>130 cm) also occurred in 2009, 2010, 2012, and 2017,
underscoring a recent pattern of higher interannual variability.  The average date of peak ice
thickness was 1-April, but was estimated to occur as early as February in some years and as late
as May in others with no trend. ITD also varied widely from 79 days in 2019 to 190 days in 2013
and averaged 144 days, also with no significant trend (Figure 8). The Kuskokwim River ice
growth curve during the winter of 2018-19 suggests 30 cm ice-thickness was reached by mid-
December and MIT was reached in late February. In contrast, during the winter of 2012-13, 30
cm ice thickness was reached in early November and peaked in early May. During the ice growth
period for the full record, air temperature at Bethel averaged -11.8 $^{0}$ C and upland snow depth
averaged 14 cm. Neither ice-growth controlling condition showed any trend during this period,
yet both air temperature and upland snow depth together explained significant portions of
variability in MIT ($r^2$=0.24, p<0.03).
Ice thickness records on the Tanana River were collected by community members
associated with the Nenana Ice Classic (NIC) and were available back to the winter of 1988-89.
In that year, MIT was 112 cm and the average for the entire record is 105 cm (Table 2). The most
recent two years had the lowest MIT on record, 69 and 72 cm, respectively—the latter, 2019, of
which was also the earliest breakup in the 102 year record, tipping the tripod on 14-April. Two
distinct ice thickness periods are noted. The earlier period from 1989 to 2007 had average MIT
of 106 cm with high interannual variability and no trend (Figure 7b). The second period showed
a strong thinning trend of 4.5 cm/yr ($r^2$=0.80, p<0.01) ending in the two thinnest ice years, as
previously noted. Ice typically thickened until early April with recorded breakup happening one
month later on average. Ice travel duration on this section of the Tanana River averaged 135 days
and ranged from 108 days in 2019 to 165 days in 2013 (Table 2) —which also corresponded to
the earliest and latest breakup dates for the total 102 year record. Ice-growth season air
temperature averaged -16.4 $^0$ C over this period and upland snow data was not consistently
available at this station over this period.

12        The second longest and mostly complete river ice observation record was made on the

Koyukuk River at Bettles starting in 1968. Three periods were noted in this record of
approximately equal durations (Figure 7c). The first was characterized by increasing thickness of
MIT by 3 cm/yr ($r^2$=0.33, p=0.02) with ice as thin as 81 cm in 1968 and as thick as 182 cm in
1978. The middle and the latest periods we identified were less distinct in terms of average MIT,
111 and 103 cm, respectively, and lacked trends, but the latter period had much higher
interannual variability with MIT ranging from 69 cm in 2009 to 182 cm in 2013 (Figure 7c). Ice
typically reached its maximum by mid- to late-April and ITD averaged 157 days with less
interannual variability than other rivers in this set (Figure 8). Ice-growth season air temperature
averaged -18.3 $^0$ C and upland snow was quite deep, averaging 54 cm and ranging from 22 cm in
1996 to 100 cm at the start of the record in 1968. Here, upland snow depth explained a
significant portion of the variation in MIT ($r^2$=0.16, p<0.05), while air temperature was
uncorrelated to ice variability over this period.

3       The Yukon River at Eagle showed an increasing trend in MIT of 1.5 cm/yr ($r^2$=0.17,

p=0.06) from 1999 to 2019, though not quite significant statistically (Figure 7d). Three distinctly
thin ice years occurred in 1999, 2004, and 2019 when the MIT reached 82 cm or less. Recent
thick ice years in 2015 and 2017, when the MIT was greater than 150 cm, contributed to this
weak trend of increasing ice growth (Figure 7d) at this eastern most station near the Canadian
border. MIT was typically reached in early April in most years and ITD ranged from 101 days in
2019 to 180 days in 2013 (Table 2).  Air temperature during the ice growth period averaged -
16.5 $^0$ C and was not correlated with variation in MIT over this period. Upland snow
observations were not consistently recorded at the station in Eagle.

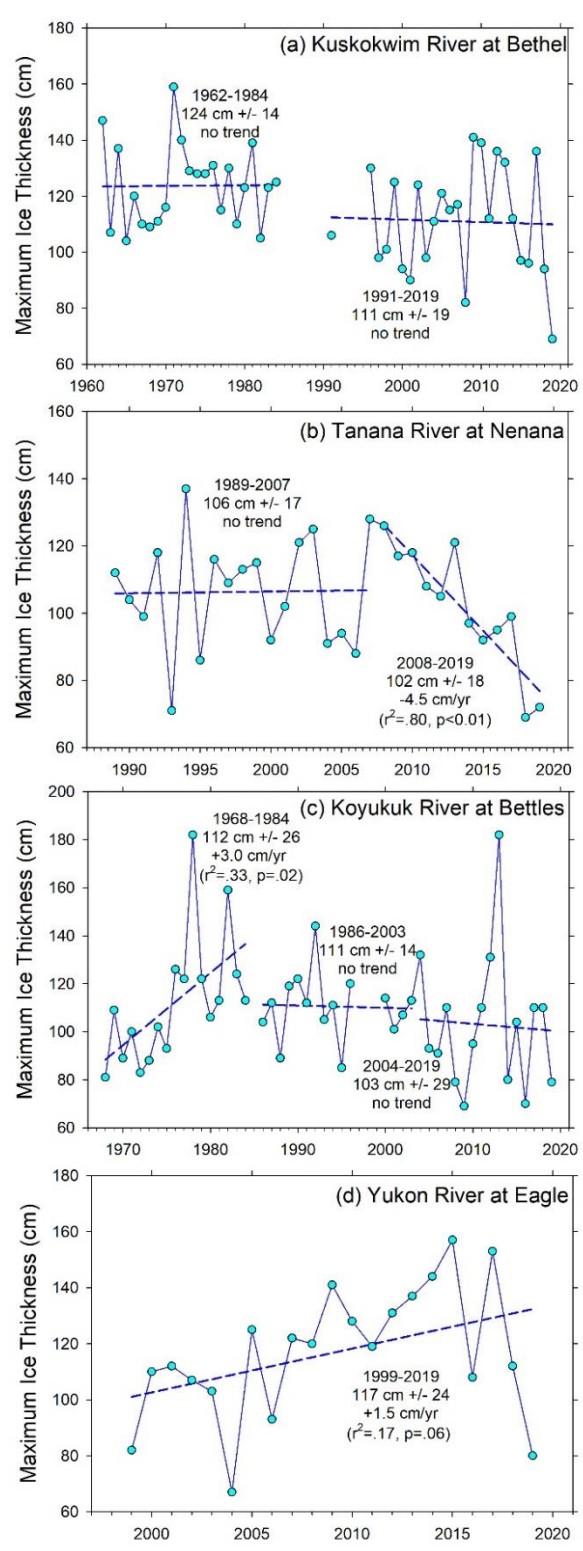

**Figure 7**. Maximum ice thickness patterns and trends for rivers near Bethel (a), Nenana (b),
Bettles (c), and Eagle (d) for each station's period of record.

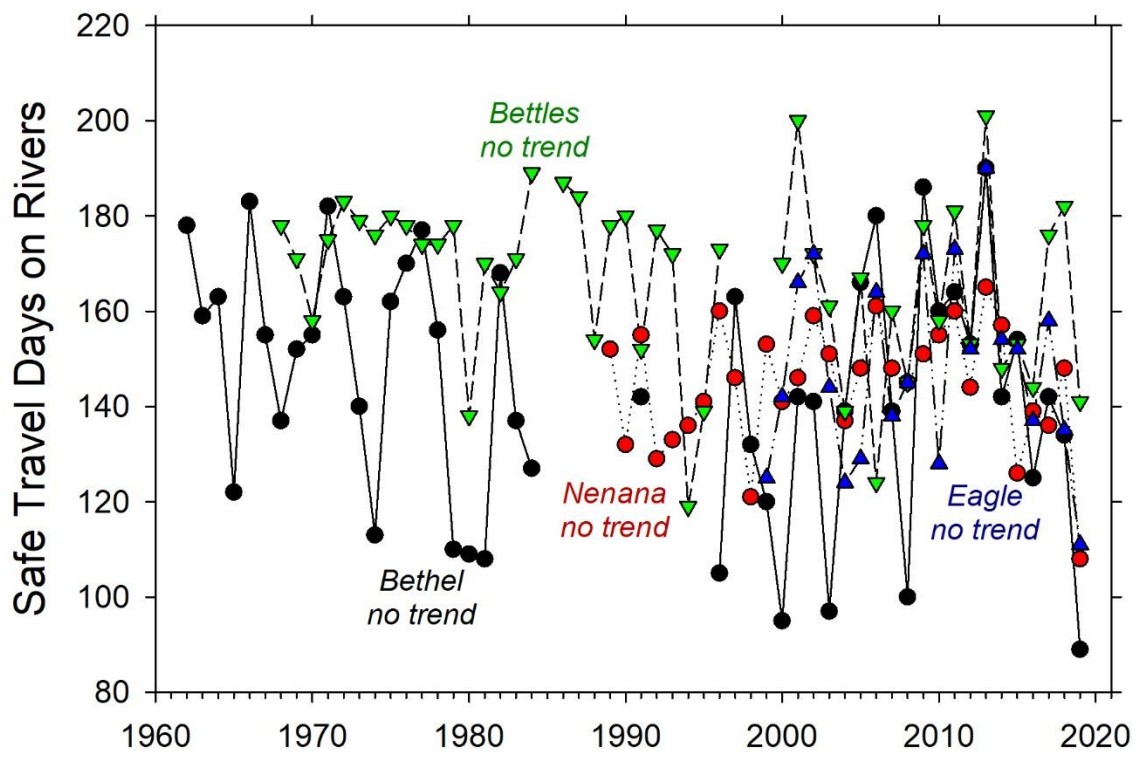

**Figure 8**. Ice travel duration on rivers estimated from ice growth curves for each period of record.

**3.3 Controls on Ice Growth**
Estimating rates of ice growth across a wide set of lakes and rivers and many years based on late
winter ice thickness observations and air temperature data produced a correspondingly wide
range of α coefficients and AFDD values (Table 1). Though not widely reported or analyzed in
ice thickness literature, α values typically range from 0.4 for snow-covered rivers to as high as
2.7 for snow-free lakes. For coastal plain lakes on the Barrow Peninsula, where we have the
widest range of variation in MIT (Figure 5a), partioning of variation using power law analysis
suggest 32% is explained by AFDD and 68% is explained by α (Figure 9). Comparison of
average α and AFDD values for all lake and river records are presented together in Figure 10a.
Here, α values for lakes in windy coastal region were all close to 2.6 with the most interannual
variability noted for Lake Hood in the southernmost site in Anchorage. The interior lakes studied
had average α values of 1.3 in Fairbanks and 2.0 in Lake Minchumina likely relating to less wind
and more consistent deep snow packs. River ice α-values were much higher than suggested in the
literature with averages ranging from 1.9 in Bettles with very deep snowpacks up to 3.3 in Bethel
where snowpacks are thinner and highly wind-affected. Though exact data on snow-ice
formation and overflow contributions to ice thickness are not consistently reported in most ice
observation data, we suspect that very high α coefficients correspond to such processes on rivers.
Analysis of factors controlling ice growth consistently point towards the dominant role of snow
in determining maximum ice thickness in most lake and river settings (Figure 10b) according to
interannual variability in thermal forcing as described by AFDD and thermal resistance as
described by α using equations 3 and 4, respectively. Analysis of Bethel and Anchorage sites,
however, show that variations in air temperature may be the more important factor for these
southernmost, coastal settings (Figure 10b).

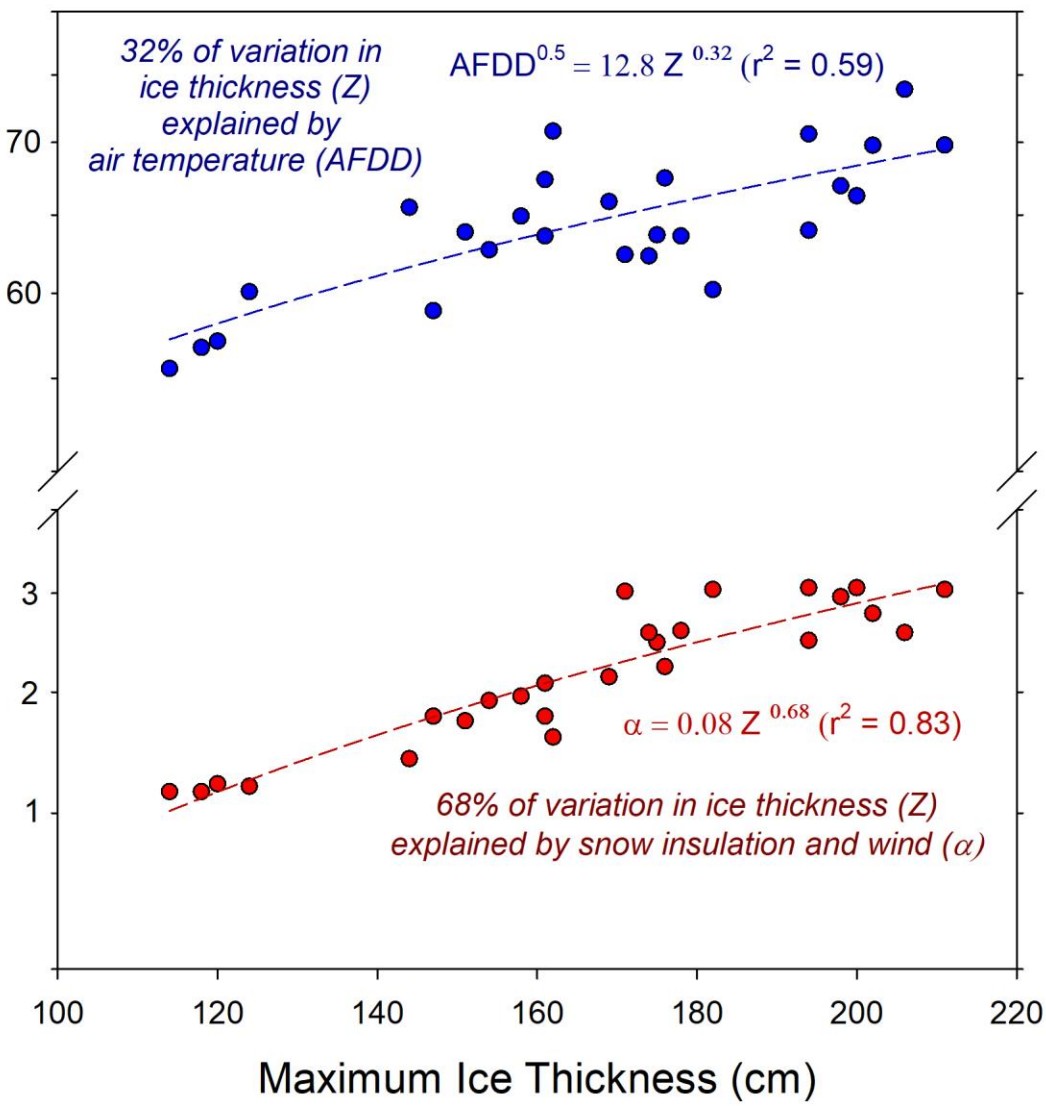

2 **Figure 9.** Example from Barrow Peninsula lake data using power law analysis partioning of

3 variation in MIT (Z) (equations 3 and 4) balanced between air temperature (AFDD) and snow

4 insulation (α).

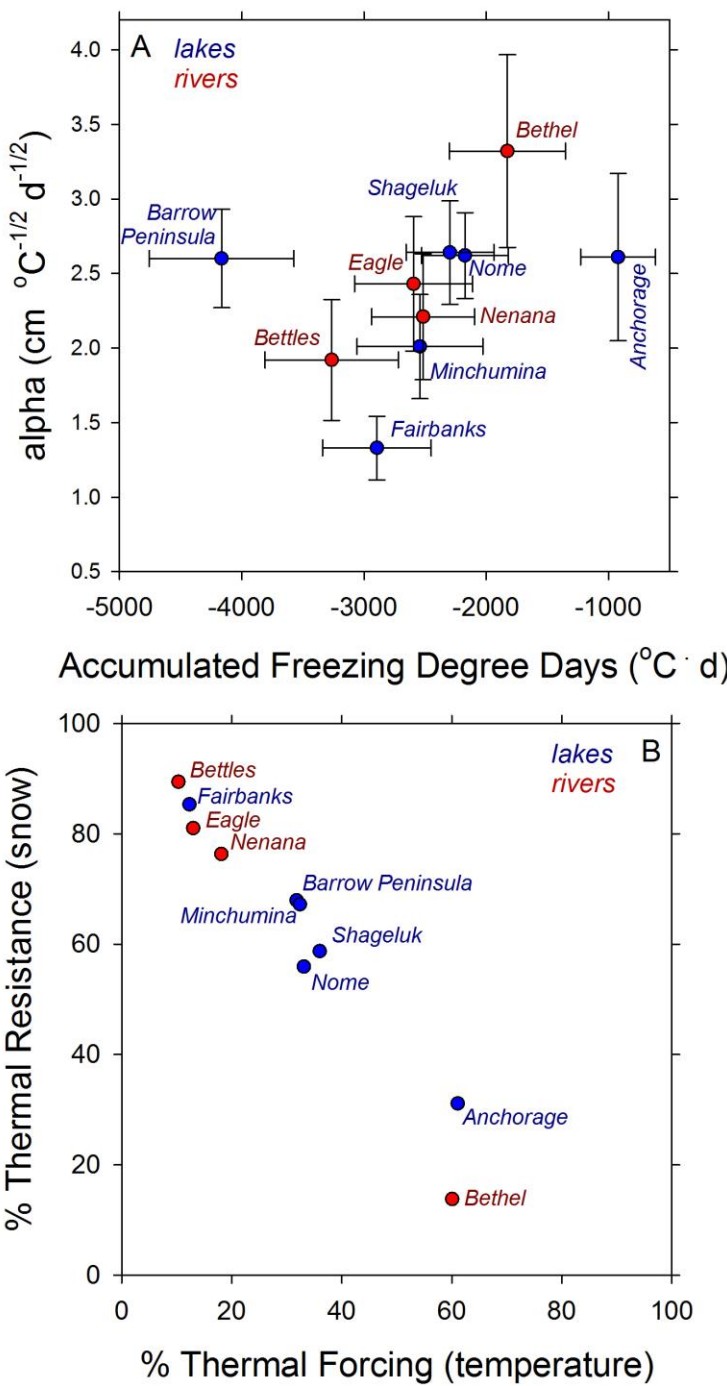

2 **Figure 10.** Comparison of mean (±1 SD) alpha and accumulated freezing degree day parameters

3 for MIT records in lakes (blue) and river (red) (A) and the proportion of variation explained by

4 thermal resistance and thermal forcing (B).

The majority of ice thickness data we report here does not have coincident measurements
of snow depth or snow density. An exception was ice thickness data collected by the CALON
project on Alaska's North Slope between 2012 and 2016, where observations of ice thickness
and snow depth and density were made in late winter close to the time of maximum ice
thickness. Short-term air temperature records collected adjacent to study lakes also enhanced the
accuracy of ice growth curve analysis and estimation of parameters. Thus, this dataset presents
an opportunity to make closer comparisons of snow characteristics to the heat exchange
coefficient $\alpha$. Increasing snow depth and decreasing snow density reduce heat loss and slow ice
growth, such that a simple Snow Insulation Index (SII) can be presented as the ratio of snow
depth to density. Comparing this SII to $\alpha$ for this North Slope MIT dataset suggest several tight
and interesting patterns (Figure 11). The combination of snow depth and density as SII explained
between 94 and 98% of variation in the heat exchange coefficient $\alpha$, but followed to distinct
separate linear relationships (Figure 11).  The steeper relationship of decreasing $\alpha$ with
increasing SII appeared to correspond to lake snowpacks of moderate depth (15 − 30 cm) and
higher densities (30 − 40 g/cc). For deeper snow and/or lower density snow, this relationship was
also tight with a shallower slope over this wider range of SII (Figure 11). One outlier
corresponded to high $\alpha$ and very low SII due to very thin and dense snow cover on a lake most
likely due to intense wind-scour. Distinction between the two linear groupings may be generally
explained by wind regimes experienced by those lakes in those years as well, though this was not
analyzed distinctly. Development of SII data for other lakes or river records were not available to
make similar comparisons.

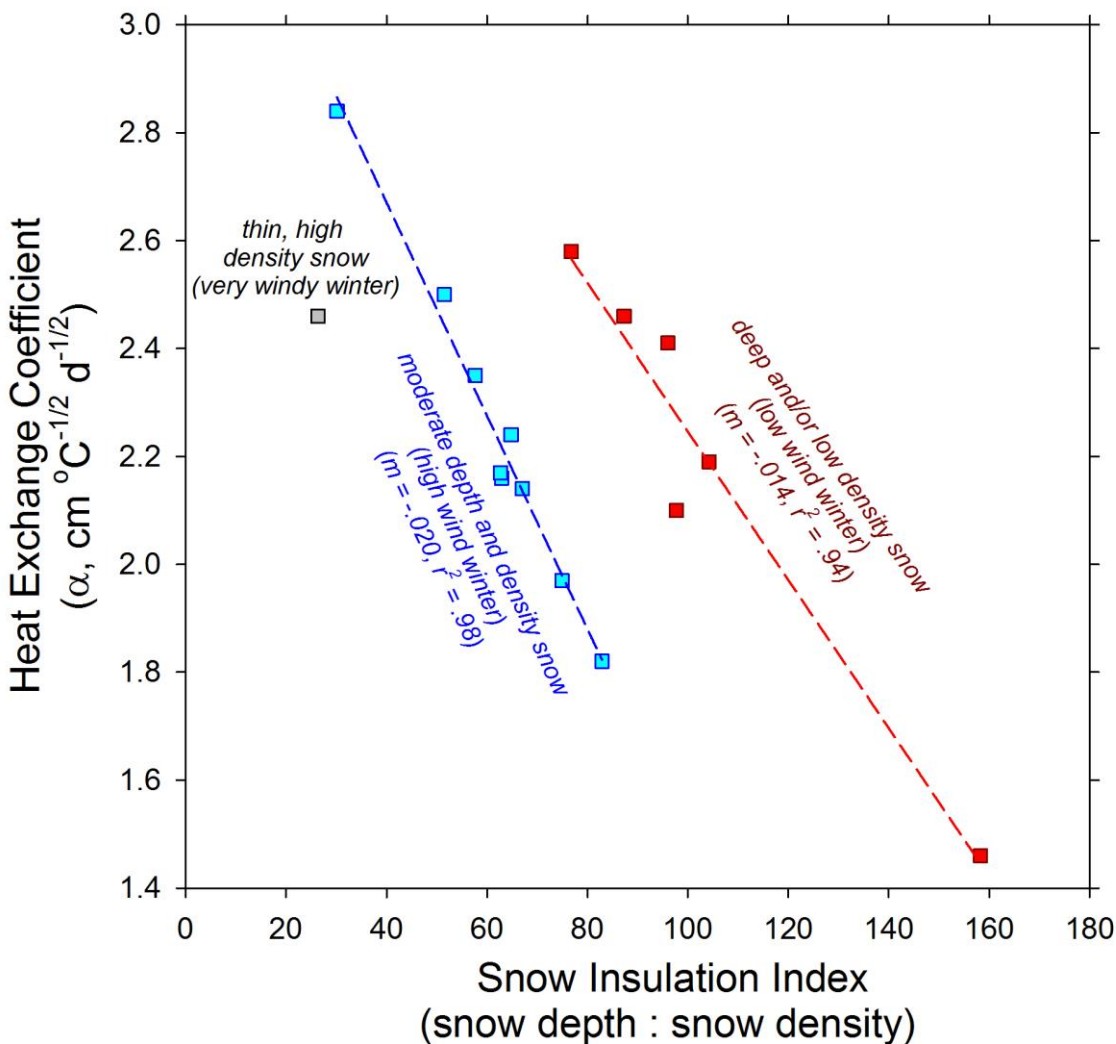

**Figure 11.** Explanation of variation in the heat exchange coefficient (α) for North Slope lake ice

near late winter MIT according to a proposed Snow Insulation Index (snow depth in cm / snow

density in g/cm$^3$). Distinct patterns emerged for snow conditions expected for low wind vs. high

wind winters, which may be applicable to other environments.

**4 Discussion**

The clearest signal of reduced ice growth and shorter duration ice cover come from lakes in northern Alaska where the impacts of arctic amplification are known to be most pronounced (Wendler et al. 2014, Walsh and Brettschneider 2019). The majority of this trend from Barrow Peninsula lakes is driven by recent declining sea ice impacts on early winter temperature and snowfall (Alexeev et al. 2016), though strong interannual variability in ice thickness is still evident (Arp et al. 2018). Barrow Peninsula ice observations from 1962 to 1996 are well within the range of more distant single year observations of 192 cm in 1955 (Brewer 1958) and 188 cm in 1882 (Ray 1885). In comparison, MIT for four of the past six years was less than 121 cm—a thickness not reported once during the previous 22 years with observations dating back to 1962. Snow is typically considered the dominant control on interannual variability in ice thickness for coastal plain lakes (Zhang and Jeffries 2000), yet warmer winter temperature during the ice growth season appear to have overcome this driver of ice growth in our analysis, at least when comparisons are made using upland snow records. Snow depth on coastal plain lakes is typically about 60% of upland depth on average (Sturm and Liston 2003), but this can vary greatly from year to year (Arp et al. 2018) and this changing offset between tundra snow and lake snowpack may explain the lack of correlation to upland snow records we observed. Partioning the relative roles of air temperature vs. snow insulation on lakes, however, still suggests that snow is the dominant factor in lake ice variability (Figure 9). The role of arctic amplification and early winter warming is also seen in reduced safe travel days on ice for Barrow Peninsula lakes (Figure 6a) with the majority of this decrease coming from slower ice thickening rather than earlier arrival of MIT.

Despite the impacts of arctic amplification on winter climate change in other parts of
Alaska (Walsh and Brettschneider 2019), the response of river and lake ice growth in several
records spanning over 50 years often appear muted or highly variable. Such variation appears
evident in analysis of the relative roles of thermal resistance due to snow and thermal forcing due
to air temperature (Figure 10a), suggesting differing process controls on ice growth across
regions and among lakes and rivers. In western coastal Alaska, recent thin ice conditions and
short ice cover duration on the Kuskokwim River were striking, yet follow a pattern of enhanced
variability over recent decades. Thin ice conditions of the 2018-19 winter were observed in
nearly all records we analyzed and provided much of our motivation to standardize, summarize,
and analyze these records. In contrast, the relatively recent winter of 2012-13 had consistently
thick ice and very prolonged ice cover duration across western and interior Alaska. Such
dramatic winter conditions and divergent ice responses underscore the need for enhanced
freshwater ice observation programs.
The premise that freshwater ice growth integrates changes in climate deserves
consideration (Allen 1977, Engram et al. 2018). We found few consistent relationships between
fundamental drivers of ice growth, air temperature and snow depth, suggesting that other more
complex environmental factors play a role in river and lake ice dynamics. One factor is that snow
accumulation on ice is fundamentally different than terrestrial upland snowpacks where snow
depth is recorded—typically ice on rivers and lakes is thinner and depending on the ice column's
isostatic balance can slow ice growth through insulation or thicken it through formation of snow-
ice and overflow (Sturm and Liston 2003, Ashton 2011). Particularly on rivers, the combined
hydrologic and thermal conditions of flowing waters can also cause divergent responses in ice
thickness—more and warmer water can cause slower growth or degradation, but also generate

overflow that can refreeze and add thickness to ice covers (Prowse and Beltaos 2002, Brown and

Duguay 2010, Jones et al. 2015). Little is known about changes in groundwater in Alaska, which

also impacts ice growth on rivers, though several studies do point to increases in groundwater

input (Brabets and Walvoord 2009, Liljedahl et al. 2017). Higher water temperatures in relation

to enhanced groundwater input present another potentially important driver affecting ice growth

and decay that deserves evaluation (Jones et al. 2015, Cherry 2019). Many of these interactions

are documented by process studies of lake and river ice (Ashton 2011) and observations of these

processes also appear sporadically in monitoring program notes (Bilello 1980), but are

challenging to quantify in long time-series analysis such as this one. Thus, ice thickness at its

seasonal maximum (MIT) and duration of the ice growth season (ITD) do integrate important

and complex changes in climate including the hydrologic cycle, but these responses do need to

be carefully interpreted and compared along with other environmental drivers.

This analysis of long-term ice observation records in Alaska using standardized metrics

from ice growth curves provides an important baseline to compare with future observations and

support process studies.  Past freshwater ice observations programs in Alaska, including CRREL

(Bilello 1980) and ALISON (Morris and Jeffries 2010), both collected basic ice thickness data

that supported numerous process studies adding to our ice dynamics predictive capability (e.g.,

Jeffries et al. 2005, Arp et al. 2010, Ashton 2011). These datasets are now archived by the Arctic

Data Center (arcticdata.io/) as Bilello (2019) and Morris and Jeffries (2019) and many of these

records continued and made readily available by APRFC

(https://www.weather.gov/aprfc/IceThickness). A new freshwater ice observation program, Fresh

Eyes on Ice (freshiceonice.org), is working to continue monitoring and analysis of river and lake

ice conditions in Alaska in part through engagement with rural communities and schools using a

combination of approaches including remote sensing and field-based observations. The strong
and interesting relationships observed between ice growth and snow characteristics for North
Slope lakes (Figure 11) may provide guidance and incentive to collect more complete snow data
to inform modeling and prediction of ice growth in other regions as well. The employment of
temperature-driven ice models that could be refined based on known or expected snow cover
conditions may provide an opportunity for near-realtime estimates or even forecasts of ice
conditions in remote regions of Alaska. Incorporation of community-based monitoring into such
efforts may not only advance more comprehensive data collection, but also promote the use of
new ice products in making safe travel decisions.
Our analysis of ice growth curves only represents a portion of the ice formation, growth,
and decay process, whereas more abundant ice phenology studies (e.g. Arp et al. 2013, Cooley
and Pavelsky 2016, Smejkalova et al. 2017, Sharma et al. 2019, Yang et al. 2020) identify
patterns and trends at the very start and very end of this cycle of seasonal ice cover. Perhaps the
two most relevant and also challenging periods are (1) the period from initial freeze-up to ice of
sufficient thickness to supporting most modes of travel (e.g. <30 cm) and (2) the period when
ice-surface snow is completely melted (when ice decay fully initiates) to break-up. These periods
in terms of process can vary greatly in length and spatial variability, both of which have great
importance for informing travel conditions and should be viewed as a continuum rather than
momentary *to-the-day* events (Brown et al. 2018). The critical freeze-up period occurs from
when ice first forms and grows on water surfaces to when ice cover is spatially consistent and
thick enough to support reliably safe travel. The critical break-up period, starts after reaching
MIT and then once snowcover is reduced to the point when ice can be exposed to direct solar
radiation and decay begins to proceed more rapidly, though often it proceeds at widely varying
rates in space and time for lakes and even more so for rivers. The importance of understanding
how these periods of freeze-up to continuous thick ice and decay initiation to break-up progress,
and how this progression may be changing over time, are critical in terms of informing safe
winter travel, predicting ice jam flood hazards, and understanding interactions with river and
lake ecosystems (Brown et al. 2018). Focusing on these key ice growth and decay periods and
how they may be responding in new ways to climate change and arctic amplification deserve
renewed attention in northern regions.
*Acknowledgements*: Numerous professional and citizen scientists contributed to ice thickness
observations used in this analysis including many dedicated volunteers from CRREL, ALISON,
and APRFC programs. Augering holes in lake and river ice is cold tedious work and their efforts
and additional observations over many years is greatly appreciated. Personally known ice
observers from Alaska's North Slope contributed hundreds of observations over recent decades
including Ben Jones, Guido Grosse, Matthew Whitman, Richard Beck, Ken Hinkel, Ben
Gaglioti, Andrew Parsekian, Andrea Creighton, and many others. This research was primarily
supported by a grant from the National Science Foundation (#1836523).

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
