# Peer review of "Observation-derived ice growth curves show patterns and trends in maximum"

_The Cryosphere, 2020_

## Referee Comment (RC1) · Anonymous Referee #1 · 27 Jun 2020

This MS presents an analysis of ice thickness data from several Alaskan lakes and rivers. The data set is wide for this topic, and the data control has been well performed. The chosen quantities to deal - max annual ice thickness (MIT) and safe travel duration (STD) - are very good to describe the role of ice thickness. The data analysis is, however, quite simple and limited, and much more information could be extracted. The MS ca The dependence of ice thickness on climate is in the first order based on air temperature and snow accumulation in lakes, in rivers also the river flow characteristics play a role. Working with freezing-degree-days (FDD) for ice growth and positive degree-days

(PDD) for ice melting is a traditional approach, but it is good to look more closely to the obtained relationships for interpretation of the role of other factors in addition to the air temperature. Here are a few comments for consideration: (1) Consider the physical interpretation of the FFD and PPD coefficients for ice phenology and thickness, See, e.g., the recent paper by Karetnikov et al. in Journal of Great Lakes Research 43(6). (2) The Stefan's formula is a good estimator of MIT although known to be biased up. The theoretical value of the coefficient a is about 3.5. If empirical fit is more, it is a very specific case. Discuss why? If the empirical fit is less, it can be explained by the snow effect but if the fit is less than 0.5*3.5 that is also a very specific case. Since you seem to have snow data, you can plot a vs. a suitable snow index to examine the role of snow. (3) Discuss ice types. How much is known about snow-ice and frail ice in your sites? Their growth does not follow Stefan formula. (4) Considering STD, Stefan formula is known to be very crude for thin as it does not limit ice-air heat fluxes. Discuss this point but using 'Stefan 30 cm' as an index for STD needs a warning. The end of STD at the time of MIT is OK but conservative. Normally lake ice is safe as long as there is snow on top since the sunlight does not deteriorate ice through snow. (5) The formula a'*PDD for ice melting is incomplete since solar radiation plays a role independently, E.g., in some Antarctic lakes ice melts with PDD almost zero. An optional formula could be e.g. melting = a"*PDD + b*(t-t0). Would this work in your sites? (6) For ice melting, the snow cover is important because snow needs to be melted first. Can you plot a' vs. snow at the time of MIT for further interpretation? (7) Considering the climate trends, the results for MIT and STD are good but it would be interesting to look into the trend of FDD for comparison and understanding where the MIT and STD trends could result.

---

## Referee Comment (RC2) · Anonymous Referee #2 · 4 Jul 2020

This work presents an excellent synthesis of a broad data set to create a comprehensive view of lake and river ice changes in Alaska. I found this to be one of the more enjoyable papers to review in quite some time and commend the authors for their well written manuscript. The length of the acquired time series are rare in ice records and highlight some important changes taking place in the regions, particularly highlighted by the comparison of trends in the long term vs recent term records. Combining the lakes and rivers into a collective freshwater ice analysis is well tied together in the discussion, especially on how the driving forces that affect ice growth/decay can both be

similar or differ.

Overall, the results this work presents tie in nicely with overall changes taking places in the cryosphere, particularly how the strongest trends were identified in the Barrow region where the sea ice and snow are also changing rapidly. It was good to see this presented in field data records. The field data presented here is one of the major strengths of this paper. Much work in the north is done with modelling and remote sensing with limited validation as field data is so challenging to come by in the remote areas. I think this manuscript presents a valuable contribution to not only Arctic ice but freshwater ice studies in general and provides further insight on an important aspect of the cryosphere. The manuscript is concise, well-illustrated, and meets the objectives. I offer a few thoughts below for the authors consideration.

Using the modified Stefan's equation, if I understand correctly, alpha and alpha prime for melt are derived by using the actual field data to fit the curves. So, in effect, even though it's a temperature driven model, you actually are capturing the effect of snow on the various ice thicknesses through that adjustment. This might be something to add a few comments on and potentially further explore. There might not be enough field data for this, but some comments on how the a and a1 values differ/compare for similar region lakes/rivers with different snow cover might lead to some interesting points. While I would normally object to temperature-only based ice models, since these curves are adjusted for actual recorded thickness measurements and the specific thickness on a certain day is not the goal (as specific snow-related thickening/melt will be missed), this modified Stefan's approach is still very valuable for an overall sense of the standardized thickness curves. The inclusion of the snow and snow-related ice processes and how the model, in some essence, captures some of these effects through the adjustment will strengthen the argument for using this approach. I do note that this approach does not work very well in southern regions where white ice is prominent (we've done it, but the results suggest some differences from observations), from what I understand of the Alaskan ice cover, the ice is predominantly thermodynamic similar to that across NWT.

A few comments on the potential limitations of excluding the non-thermodynamic ice in the more southern regions of Alaska (where I am not familiar with the ice - perhaps this region still has limited white ice and hence this is not relevant) might also strengthen the rational for the methodology.

The margin of error for ice thickness measurements is a great addition. I like this very much and can see this being quite useful for researchers in regions that do not have large/long datasets such as that in Alaska to get a sense of the utility of their samples.

I also quite like the ITD metric derived for analysis as well. Having the 'safe to travel' data for comparison is a tremendous asset that lends confidence to this, albeit conservative, method for examining the safe travel duration. I think your ITD metric will also be quite useful for ice researchers to learn from and modify to their respective regions.

In your results section where you examine if the temperature or snow explained the variations in the ice thickness (pg. 18-22), have you explored the combined effects somehow? Results we come across more frequently now in our work (in more southern regions) is that it appears snowfall changes are buffering the temperature changes with respect to their effects on the ice thickness. I do not disagree at all with the discussion of these results pointing out that other drivers are likely in play here, but perhaps a few comments on the combined effects of temperature/snow changes might offer some explanation for the lakes that show no agreement.

Very small suggestion here, define (or refresh the reader) what is meant by upland for the snow measurements in the results. I intuitively think 'upland' somewhere in the basin, though I understand this be on shore at the nearest weather station.

On page 19, the comments about the observers potentially taking measurements in the compact snow vs. undisturbed snow is an excellent point to highlight. I think this warrants an additional sentence on how the compaction may have led to further thickening. That might be a beneficial clarification for researchers who do not focus on lake ice.

I was pleased to read a good discussion of the variability in snow depths between upland regions and on the ice. Some good points were raised here, while the average on-ice snow depth is 60% (in Alaska) of that on-land the annual variability can be very large. Are there any on-ice snow measurement you can use to further explore this on any of your study lakes that don't have good correlation to snow upland? Particular geographic settings leading to much larger/smaller percentages on ice? This may be beyond the purview of the existing dataset, but just a thought to explore if the data exists.

Technical comments:

Pg 2, Line 9 – Serreze and Francis 2006 – that's quite out of date for Arctic research, I suggest using a more up to date reference.

Pg 3, line 7 – I believe rivers should be plural

Pg 8, line 18 – TWITS. I absolutely love this.

Pg 28, line 4 – I believe temperature should be plural

---

## Author Comment (AC1) · 26 Aug 2020

Dear Christian Haas,

Please see our replies to the two anonymous reviewer comments as well as a track-changes version of our manuscript entitled "Observation-derived ice growth curves show patterns and trends in maximum ice thickness and safe travel duration of Alaskan lakes and rivers". We have addressed all comments and requests brought up by these reviewers and hope that the revised version is acceptable for publication in The Cryosphere.

Kind regards,

Chris Arp (on behalf of all authors)

Dear Reviewer (RC1),

We thank you very much for your comments which will help to increase the quality of our manuscript. Please find in the following a point-by-point reply to your review. We furthermore provide the revised version of our manuscript as well as a track-changes version in which individual changes with respect to the submitted manuscript are highlighted. Your comments are in **bold,** Responses: in plain text, and extracts from the manuscript are in *italics*.

Anonymous Referee #1

**This MS presents an analysis of ice thickness data from several Alaskan lakes and rivers. The data set is wide for this topic, and the data control has been well performed. The chosen quantities to deal - max annual ice thickness (MIT) and safe travel duration (STD) - are very good to describe the role of ice thickness. The data analysis is, however, quite simple and limited, and much more information could be extracted. The dependence of ice thickness on climate is in the first order based on air temperature and snow accumulation in lakes, in rivers also the river flow characteristics play a role. Working with freezing-degree-days (FDD) for ice growth and positive degree-days (PDD) for ice melting is a traditional approach, but it is good to look more closely to the obtained relationships for interpretation of the role of other factors in addition to the air temperature. Here are a few comments for consideration: (1) Consider the physical interpretation of the FFD and PPD coefficients for ice phenology and thickness, See, e.g., the recent paper by Karetnikov et al. in Journal of Great Lakes Research 43(6).**

Response: We have downloaded and studied this manuscript by Karentnikov et al. and find it quite fascinating in terms of the region and definitely the temporal extent. I think we have done a better job with interpretation of FDD in the additional analysis described below. However we have not found a reasonable way to incorporate directly the type of analysis described in the Karetnikov paper.

**(2) The Stefan's formula is a good estimator of MIT although known to be biased up. The theoretical value of the coefficient a is about 3.5. If empirical fit is more, it is a very specific case. Discuss why? If the empirical fit is less, it can be explained by the snow effect but if the fit is less than 0.5*3.5 that is also a very specific case. Since you seem to have snow data, you can plot a vs. a suitable snow index to examine the role of snow.**

Response: These are great points regarding interpretation of the alpha coefficient, though we haven't been able to find exact reference that discuss these relationship (i.e. a > 3.5 and a < 1.75). In a much earlier draft of this manuscript, we actually went into

much more analytical detail of how alpha related to snow and in fact developed a Snow Insulations Index based on snow depth relative to density. Based on the reviewers comments we have now added some of this back into the manuscript as a results section (p 28-33):

**3.3 Controls on Ice Growth**

*Estimating rates of ice growth across a wide set of lakes and rivers and many years based on late winter ice thickness observations and air temperature data produced a correspondingly wide range of α coefficients and AFDD values (Table 1). Though not widely reported or analyzed in ice thickness literature, α values typically range from 0.4 for snow-covered rivers to as high as 2.7 for snow-free lakes. For coastal plain lakes on the Barrow Peninsula, where we have the widest range of variation in MIT (Figure 5a), partioning of variation using power law analysis suggest 32% is explained by AFDD and 68% is explained by α (Figure 9). Comparison of average α and AFDD values for all lake and river records are presented together in Figure 10a. Here, α values for lakes in windy coastal region were all close to 2.6 with the most interannual variability noted for Lake Hood in the southernmost site in Anchorage. The interior lakes studied had average α values of 1.3 in Fairbanks and 2.0 in Lake Minchumina likely relating to less wind and more consistent deep snow packs. River ice α-values were much higher than suggested in the literature with averages ranging from 1.9 in Bettles with very deep snowpacks up to 3.3 in Bethel where snowpacks are thinner and highly wind-affected. Though exact data on snow-ice formation and overflow contributions to ice thickness are not consistently reported in most ice observation data, we suspect that very high α coefficients correspond to such processes on rivers. Analysis of factors controlling ice growth consistently point towards the dominant role of snow in determining maximum ice thickness in most lake and river settings (Figure 10b) according to interannual variability in thermal forcing as described by AFDD and thermal resistance as described by α using equations 3 and 4, respectively. Analysis of southernmost coastal site of Bethel and Anchorage, however, show that variation in air temperature may be the more important factor (Figure 10b).*

[Figure]

**Figure 9.** *Example from Barrow Peninsula lake data using power law analysis partioning of variation in MIT (Z) (equations 3 and 4) balanced between air temperature (AFDD) and snow insulation (α).*

[Figure]

**Figure 10.** *Comparison mean (±1 SD) alpha and accumulated freezing degree day parameters for MIT records in lakes (blue) and river (red) (A) and the proportion of variation explained by thermal resistance and thermal forcing (B).*

The majority of ice thickness data we report here does not have consident measurements of on ice snow depth and information on snow density is even more rarely collected on a consistent basis. An exception was ice thickness data collected by the CALON project on Alaska's North Slope between 2012 and 2016, where observations of ice thickness and snow

*depth and density were made in late winter close to the time of maximum ice thickness. Short-term air temperature records collected close by study lakes also enhanced accuracy of ice growth curve analysis and estimation of parameters. Thus, this dataset presents an opportunity to make closer comparisons of snow characteristics to the heat exchange coefficient α. Increasing snow depth and decreasing snow density reduce of heat loss and slow ice growth, such that a simple Snow Insulation Index (SII) can be presented as the ratio of snow depth to density. Comparing this SII to α for this North Slope MIT dataset suggest several tight and interesting patterns (Figure 11). The combination of snow depth and density as SII explained between 94 and 98% of variation in the heat exchange α, but followed to distinct linear relationships (Figure 11). The steeper relationship of decreasing α with increasing SII appeared to correspond to lake snowpacks of moderate depth (15 – 30 cm) and higher densities (30 – 40 g/cc). For deeper snow and/or lower density snow, this relationship was also tight with a shallow slope over this wider range of SII (Figure 11). One outlier corresponded to high α and very low SII due to very thin and dense snow cover on a lake most likely due to intense wind-scour. Distinction between the two linear grouping may be explained by wind regimes experienced by those lakes in those years as well, though this was not analyzed distinctly. Development of SII data for other lakes or river records were not available to make similar comparisons.*

[Figure]

**Figure 11.** *Explanation of variation in the heat exchange coefficient (α) for North Slope lake ice near late winter MIT according to a proposed Snow Insulation Index (snow depth in cm / snow density in g/cm³). Distinct patterns emerged for snow conditions expected for low wind vs. high wind winters, which may be applicable to other environments.*

We think the reviewer will find this of particular interest and we are delighted that they brought this up as we were concerned about this level of details and relative to other components of the manuscript, but based on the reviewers comments are very happy to add this back in.

**(3) Discuss ice types. How much is known about snow-ice and frail ice in your sites? Their growth does not follow Stefan formula.**

Response: We have now added additional discussion of the potential role of snow-ice in the manuscript, which does in fact deviate from expected alpha values of the Stefan equation (p 29 ln 9-11):

*Though exact data on snow-ice formation and overflow contributions to ice thickness are not consistently reported in most ice observation data, we suspect that very high α coefficients correspond to such processes on rivers.*

**(4) Considering STD, Stefan formula is known to be very crude for thin as it does not limit ice-air heat fluxes. Discuss this point but using 'Stefan 30 cm' as an index for STD needs a warning. The end of STD at the time of MIT is OK but conservative. Normally lake ice is safe as long as there is snow on top since the sunlight does not deteriorate ice through snow.**

Response: This point is definitely valid, which we thought was included in the manuscript but could not find it (may have been edited out). In any event, we have now included this important detail which helps us justify selecting 30 cm as the start of STD. Sentenced included now is (p13 ln 16-18):

*We also note that modeling ice growth during the initial thickening phase is less predictable by air temperature relationship (Ashton 1989) and thus selected a level of thickness where we expect this relationship to be more robust.*

Response: We agree that using the time of MIT is conservative and did state this in the original text with justification of not having sufficient data on decay. Original text was (p14 ln 5-7):

*Thus, our estimates of ITD should be considered conservative and for many modes of travel (and corresponding levels of caution) our estimates of ITD are shorter in duration that what is practiced locally.*

Response: We have also now added additional explanation in the discussion regarding snow cover and sunlight impacts on ice decay, which reads (p 37 ln 21-23):

*The critical break-up period, starts after reaching MIT and then once snowcover is reduced to the point when ice can be exposed to direct solar radiation and decay begins to proceed more rapidly, though often it proceeds at widely varying rates in space and time for lakes and even more so for rivers.*

**(5) The formula a'*PDD for ice melting is incomplete since solar radiation plays a role independently, E.g., in some Antarctic lakes ice melts with PDD almost zero. An optional formula could be e.g. melting = a''*PDD + b*(t-t0). Would this work in your sites?**

Response: We wish we had data to evaluate this comprehensively across a wide number of lakes. We do have some solar radiation data specific to lakes, but these are very limited in time and space and would not fit well within this analysis. We actually do have plans to analyze these data at some point and work on developing the relationships as the reviewer proposes.

(**6) For ice melting, the snow cover is important because snow needs to be melted first. Can you plot a' vs. snow at the time of MIT for further interpretation?**

Response: Even though we use alpha prime in our ice curve model, we have much sparser data during the decay phase to realistically fit the decay period, which is one reason we have not included this in the analysis and only analyze the growth portion of the curve in this paper. Again this is a very interesting point and one we'd like to follow up on at sites where we have this data.

**(7) Considering the climate trends, the results for MIT and STD are good but it would be interesting to look into the trend of FDD for comparison and understanding where the MIT and STD trends could result.**

Response: I believe the reviewer is asking us to analyze FDD trends alone, which if this is the case is certainly interesting, but we think has generally been done more abundantly by others in the arctic climate science community and might be too much for the scope of this paper. We do now add comparisons of FDD to MIT as in Figure 9 and Figure 10, which is a good addition and we appreciate the reviewer making this suggestion.

Dear Reviewer (RC2),

We thank you very much for your comments which will help to increase the quality of our manuscript. Please find in the following a point-by-point reply to your review. We furthermore provide the revised version of our manuscript as well as a track-changes version in which individual changes with respect to the submitted manuscript are highlighted. Your comments are in **bold,** Responses: in plain text, and extracts from the manuscript are in *italics*.

**This work presents an excellent synthesis of a broad data set to create a comprehensive view of lake and river ice changes in Alaska. I found this to be one of the more enjoyable papers to review in quite some time and commend the authors for their well written manuscript. The length of the acquired time series are rare in ice records and highlight some important changes taking place in the regions, particularly highlighted by the comparison of trends in the long term vs recent term records. Combining the lakes and rivers into a collective freshwater ice analysis is well tied together in the discussion, especially on how the driving forces that affect ice growth/decay can both be similar or differ.**

Response: We greatly appreciate the reviewers encouraging and thoughtful comments here.

**Overall, the results this work presents tie in nicely with overall changes taking places in the cryosphere, particularly how the strongest trends were identified in the Barrow region where the sea ice and snow are also changing rapidly. It was good to see this presented in field data records. The field data presented here is one of the major strengths of this paper. Much work in the north is done with modelling and remote sensing with limited validation as field data is so challenging to come by in the remote areas. I think this manuscript presents a valuable contribution to not only Arctic ice but freshwater ice studies in general and provides further insight on an important aspect of the cryosphere. The manuscript is concise, well-illustrated, and meets the objectives. I offer a few thoughts below for the author's consideration. Using the modified Stefan's equation, if I understand correctly, alpha and alpha prime for melt are derived by using the actual field data to fit the curves. So, in effect, even though it's a temperature driven model, you actually are capturing the effect of snow on the various ice thicknesses through that adjustment. This might be something to add a few comments on and potentially further explore.**

Response: Yes, this is definitely the case that alpha represents the impact of snow and likely other environmental drivers. Based on this comment and those of the previous reviewer on this topic we have added a new section to the results (p 28 ln 5):

*3.3 Controls on Ice Growth*

**There might not be enough field data for this, but some comments on how the a and a1 values differ/compare for similar region lakes/rivers with different snow cover might lead to some interesting points. While I would normally object to temperature-only based ice models, since these curves are adjusted for actual recorded thickness measurements and the specific thickness on a certain day is not the goal (as specific snow-related thickening/melt will be missed), this modified Stefan's approach is still very valuable for an overall sense of the standardized thickness curves. The inclusion of the snow and snow-related ice processes and how the model, in some essence, captures some of these effects through the adjustment will strengthen the argument for using this approach. I do note that this approach does not work very well in southern regions where white ice is prominent (we've done it, but the results suggest some differences from observations), from what I understand of the Alaskan ice cover, the ice is predominantly thermodynamic similar to that across NWT.**

Response: This point is quite interesting and matches new results we present for our more southerly sites (Anchorage and Bethel), which are now shown in Figure 10b.

**I was pleased to read a good discussion of the variability in snow depths between upland regions and on the ice. Some good points were raised here, while the average on-ice snow depth is 60% (in Alaska) of that on-land the annual variability can be very large. Are there any on-ice snow measurement you can use to further explore this on any of your study lakes that don't have good correlation to snow upland? Particular geographic settings leading to much larger/smaller percentages on ice? This may be beyond the purview of the existing dataset, but just a thought to explore if the data exists.**

Response: There are some, but the only sites were we have sufficiently comprehensive data is on the North Slope and have now included this lake snow data in Figure 11. We do make a point of this need in the Discussion now, which reads (p 37, ln 1-9):

*The strong and interesting relationships observed between ice growth and snow characteristics for North Slope lakes (Figure 11) may provide guidance and incentive to collect more complete snow data to inform modeling and prediction of ice growth in other regions as well. The employment of temperature-driven ice models that could be refined based on known or expected snow cover conditions may provide an opportunity for near-realtime estimates or even forecasts of ice conditions in remote regions of Alaska. Incorporation of community-based monitoring into such efforts may not only advance more comprehensive data collection, but also use of new ice products in making safe travel decisions.*

**Technical comments:**

**Pg 2, Line 9 – Serreze and Francis 2006 – that's quite out of date for Arctic research, I suggest using a more up to date reference.**

Response: We agree that this is an older reference, but are not aware of a reference stating this exact point more recently. It's very likely there is, but we just have not found it.

**Pg 3, line 7 – I believe rivers should be plural**

Response: Corrected, thanks for seeing this.

**Pg 8, line 18 – TWITS. I absolutely love this.**

Response: Yes, it is cool. Thank Kim Morris and Martin Jeffries for that!

**Pg 28, line 4 – I believe temperature should be plural**

Response: Corrected, thanks for seeing this.

[revised manuscript text omitted]